# Real-World Dispensing of Buprenorphine in California during Prepandemic and Pandemic Periods

**DOI:** 10.3390/healthcare12020241

**Published:** 2024-01-18

**Authors:** Yun Wang, Alexandre Chan, Richard Beuttler, Marc L. Fleming, Todd Schneberk, Michael Nichol, Haibing Lu

**Affiliations:** 1School of Pharmacy, Chapman University, Irvine, CA 92618, USA; rbeuttle@chapman.edu (R.B.); mfleming@chapman.edu (M.L.F.); 2Department of Clinical Pharmacy Practice, School of Pharmacy & Pharmaceutical Sciences, University of California Irvine, Irvine, CA 92697, USA; a.chan@uci.edu; 3Gehr Center for Health Systems Science and Innovation, Keck School of Medicine, University of Southern California, Los Angeles, CA 90033, USA; todd.schneberk@med.usc.edu; 4Sol Price School of Public Policy, University of Southern California, Los Angeles, CA 90089, USA; mnichol@usc.edu; 5Leavey School of Business, Santa Clara University, Santa Clara, CA 95053, USA; hlu@scu.edu

**Keywords:** buprenorphine, opioid use disorder, opioids, healthcare access, COVID-19

## Abstract

Introduction: The opioid overdose crisis in the United States has become a significant national emergency. Buprenorphine, a primary medication for individuals coping with opioid use disorder (OUD), presents promising pharmacokinetic properties for use in primary care settings, and is often delivered as a take-home therapy. The COVID-19 pandemic exacerbated the scarcity of access to buprenorphine, leading to dire consequences for those with OUD. Most existing studies, primarily focused on the immediate aftermath of the COVID-19 outbreak, highlight the challenges in accessing medications for opioid use disorder (MOUDs), particularly buprenorphine. However, these studies only cover a relatively short timeframe. Methods: To bridge this research gap, in our study, we utilized 33 months of California’s prescription drug monitoring program (PDMP) data to provide insights into real-world buprenorphine dispensing trends since the onset of the pandemic from 2018 to 2021, focusing on outcomes such as patient counts, prescription volumes, prescriber involvement, days’ supply, and dosage. Statistical analysis employed interrupted time series analysis to measure changes in trends before and during the pandemic. Results: We found no significant impact on patient counts or prescription volumes during the pandemic, although it impeded the upward trajectory of prescriber numbers that was evident prior to the onset of the pandemic. An immediate increase in days’ supply per prescription was observed post-pandemic. Conclusion: Our findings differ in comparison to previous data regarding the raw monthly count of patients and prescriptions. The analysis encompassed uninsured patients, offering a comprehensive perspective on buprenorphine prescribing in California. Our study’s insights contribute to understanding the impact of COVID-19 on buprenorphine access, emphasizing the need for policy adjustments.

## 1. Introduction

Opioid overdose has escalated to a critical national crisis in the United States. Between 1999 and 2019, approximately 500,000 lives were lost to overdoses involving prescription and illicit opioids [1]. The onset of the COVID-19 pandemic ushered in a surge in drug overdose fatalities [2,3], with over 75% of these tragic events involving opioids [4]. Opioid overdose deaths remain at crisis levels during the post-coronavirus disease (COVID-19) pandemic era [4]. In California alone, drug overdoses claimed the lives of 10,000 individuals in the year ending in September 2021, marking a staggering 70% increase from 2019 [5].

Approximately 1 in 4 patients receiving opioid prescriptions for pain management develops opioid use disorder (OUD) [6]. Effective medications for opioid use disorder (MOUDs) are one reason for optimism [7,8]. However, MOUDs are underutilized [9,10,11,12,13,14,15]. Several individual characteristics put patients in a vulnerable position in respect of their access to MOUD treatments; for example, racial/ethnic and socioeconomic disparities, mental disease [16], and certain types of insurance (e.g., Medicare) [17,18]. Neighborhood characteristics also impede patients’ access to MOUDs [9,11,12,13,14,19]. The racial composition of neighborhoods has led to a disparity in the supply of first-line MOUDs (i.e., buprenorphine and methadone) [20]. Extensive prior research has identified substantial disparities between the needs of patients and their access to buprenorphine treatment. Only 1 in 4 people who needed treatment received MOUDs [21]. A disconcerting trend has emerged as patients reported that 38–46% of their appointments were declined by buprenorphine prescribers, primarily due to their being new, uninsured, self-paying patients or Medicaid beneficiaries [9]. Clinicians, however, have cited an array of barriers, including limited access to addiction specialists and behavioral health services, reluctance to prescribe, skepticism regarding agonist treatment, reimbursement challenges, and inadequate institutional support (e.g., insufficient backing for providing medication-assisted treatment (MAT) and time constraints) [9,11,12,13,14].

Buprenorphine, the frontline medication, presents a range of favorable pharmacokinetic traits that make it well-suited for utilization in primary care settings when compared to methadone [22]. It is administered as a take-home therapy, offering a practical solution for individuals dealing with OUD. The advent of the COVID-19 pandemic further exacerbated the scarcity of access to buprenorphine, resulting in dire consequences for individuals with OUD. The prevailing literature from various states has emphasized the intricate challenges posed by the COVID-19 pandemic in respect of accessing MOUDs, with a particular focus on buprenorphine [23,24,25,26,27]. To address this pressing issue, California has made substantial investments in the medication-assisted treatment (MAT) expansion project to enhance access to OUD treatment with buprenorphine or methadone [5].

As the most populous state in the nation, California faces an urgent imperative to address the health disparities experienced by patients in dire need of buprenorphine treatments. However, most studies [23,24,25,26,27] have primarily focused on the immediate aftermath of the outbreak within a relatively short timeframe. In contrast, in our study, we harnessed 33 months of California’s prescription drug monitoring program (PDMP) data from the Controlled Substance Utilization Review and Evaluation System (CURES) to bridge the existing research gap and provide insights into real-world buprenorphine dispensing trends since the onset of the COVID-19 pandemic.

## 2. Materials and Methods

We utilized data from the CURES, covering the period 2018 to 2021. CURES serves as a state-operated PDMP repository that consolidates information regarding Schedule II-V prescription drugs dispensed by outpatient pharmacies within California. Notably, buprenorphine, classified as a controlled Schedule III drug, is encompassed within the CURES database. A comprehensive description of this database can be found in our prior studies [28,29] and on the official webpage of the California Department of Justice [30]. The existing body of literature in other states has underscored the multifaceted challenges introduced by the COVID-19 pandemic in respect of accessing medications for opioid use disorder (MOUDs), particularly buprenorphine. The dataset encompasses various variables, including patient attributes such as year of birth and sex, patient residence details (city, state, and 5-digit zip code), and corresponding prescriber and pharmacy information (5-digit zip code). Additionally, it provides specifics about the prescription itself, including product name, National Drug Code (NDC), formulation, strength, and quantity.

### 2.1. Procedure

We delineated the “pre-pandemic period” as the year spanning from 19 March 2019 to 18 March 2020, which coincided with the commencement of the statewide stay-at-home order. Subsequently, the “pandemic period”, encompassing 21 months, extended from 19 March 2020 to 18 December 2021. We assessed various outcomes every month during both these periods, which included (1) the monthly volume of patients filling buprenorphine prescriptions, (2) the monthly volume of buprenorphine prescription refills, (3) the monthly volume of prescribers involved, (4) the mean days’ supply per prescription, and (5) the mean daily dosage among patients filling buprenorphine prescriptions each month. To maintain result accuracy, we excluded certain buprenorphine product formulations, such as extended-release solutions, implants, powders, and those labeled for analgesic purposes. Daily buprenorphine doses range from 8 mg to a maximum of 24 to 32 mg [31,32]. We excluded prescriptions with excessive daily amounts exceeding 50 mg.

### 2.2. Data Analysis

For our statistical analysis, we employed interrupted time series analysis, a widely recognized and robust approach to quantify changes in trends before and during the COVID-19 pandemic. Interrupted time series [33] is a potent quasi-experimental research design and a powerful tool for evaluating the impact of policy changes or quality improvement programs on the rate of a specific outcome in a defined population. The time series, which involves repeated observations of an event over time, is divided into two segments. The first segment encompasses rates of the event before the intervention or policy, marked in this case by the onset of the COVID-19 outbreak in March 2020, while the second segment represents rates after March 2020. “Segmented regression” was employed to statistically measure changes in level and slope during the post-pandemic period compared to the pre-pandemic period. The term “segmented” indicates a model with different intercept and slope coefficients for the pre- and post-pandemic time periods [33]. Segmented regression helps assess immediate (level) changes in the outcome rate and trend (slope). In standard interrupted time series analyses [34], the following segmented regression model is employed: *Yt* = *β*_0_ + *β*_1_*T* + *β*_2_*X_t_* + *β*_3_*TX_t_*, where *β*_0_ represents the baseline level at *T* = 0, *β*_1_ is interpreted as the change in the outcome associated with a time unit increase (representing the underlying pre-intervention trend), *β*_2_ is the level change following the intervention, and *β*_3_ indicates the slope change following the intervention (utilizing the interaction between time and intervention *TX_t_*). We report the significance levels of the *β*_1_ (baseline level), *β*_2_ (level change), and *β*_3_ (the slope change) coefficients. A positive *β*_1_ value suggests an upward trend, while a negative value indicates a declining tendency.

We analyzed data using Stata SE, version 17 (StataCorp LLC, College Station, TX, USA), with a predetermined significance level of 0.05. The Institutional Review Board at Chapman University granted an exemption to review the study due to the deidentified nature of the data.

## 3. Results

In the twelve months leading up to the pandemic, a total of 92,723 patients received 640,883 buprenorphine prescriptions. In the subsequent 21 months following March 2020, an impressive 983,961 buprenorphine prescriptions were dispensed, benefiting 126,957 patients (refer to Table 1). The mean supply duration per prescription stood at 20.67 days before the onset of the pandemic and increased to 22.10 days after May 2020.

Pre-pandemic, an average of 3916 prescribers actively prescribed buprenorphine monthly. However, following March 2020, this monthly figure increased to 4227. The growth rate before the pandemic demonstrated a significant monthly increase of 53 prescribers (*p* < 0.001; see Figure 1, Appendix A, and Table 2). Unfortunately, the upward trajectory was disrupted by the outbreak of COVID-19 (*p* < 0.001; see Figure 1, Appendix A, and Table 2). Post-March 2020, an immediate increase was observed in the mean days of supply per refill, with a monthly rise of 1 day (*p* < 0.001; see Appendix A and Table 2). Pre-pandemic, prescribers experienced a notable increase in dosage (*p* = 0.001; see Appendix A and Table 2), and while this trend appears to have slightly diminished since the pandemic, it is not statistically significant (see Appendix A and Table 2). Despite the pandemic’s impact on moderating the growth of active prescribers, there was no notable change in patient numbers or monthly prescription volumes (refer to Figure 1, Appendix A and Table 2).

## 4. Discussion

Throughout the 21 months following California’s statewide stay-at-home order in response to COVID-19, we did not identify any significant pandemic-related impact on the trend changes of monthly patient counts or prescription volumes. However, the study revealed a moderation in the previously upward trend in the number of prescribers. In March 2020, there was a sudden increase in the number of days’ supply per prescription, though no changes were observed in daily dosage or days of supply over the extended period.

In comparison, recent IQVIA claims data from Symphony Health [35] suggest that while there was an initial increase in patients prescribed buprenorphine during the early stages of the pandemic, the monthly rate of patients prescribed buprenorphine increased at a slower pace compared to the pre-pandemic period. Furthermore, there was a decline in the number of buprenorphine prescriptions dispensed, both in terms of quantity and growth rate, during the pandemic [35]. An increase in the average number of days’ supply of buprenorphine prescriptions was also identified [35]. Although our study in California revealed an erosion in the raw monthly count of patients filling buprenorphine prescriptions and a reduction in prescriptions, our findings indicate that these changes occurred without a noticeable alteration in the overall trend. Interestingly, both studies showed an increase in the average number of days’ supply of buprenorphine prescriptions, with our findings indicating an immediate rise following March 2021. Compared to claims data, which exclusively encompass insured patients, our state-level PDMP database covered a broader spectrum of patients representing diverse insurance statuses. A survey focused on callers seeking buprenorphine treatment for OUD in several states highlighted that many buprenorphine prescribers do not accept Medicaid or uninsured patients [36]. In 2022, an estimated 3.2 million individuals in California remained uninsured [37]. The inclusiveness of Medicaid and uninsured patients in state-level PDMP data could facilitate understanding access to MOUDs for all patients under multiple health insurance statutes and better inform policymakers. Our analysis provides a comprehensive perspective on buprenorphine prescribing in California during the pandemic as the state PDMP program collected all the dispensing records of buprenorphine from all the pharmacies in California.

The gradual increase in active buprenorphine prescribers in our observations was a positive sign before the pandemic. Previous literature suggested that buprenorphine prescriptions and prescribers increased by 36% and 86% from 2016 to 2021 nationally [38]. During the pandemic, the Drug Enforcement Administration temporarily relaxed outpatient buprenorphine prescribing regulations [27,39]. Revisions included permitting prescribing to new patients via telephone or telemedicine and to existing patients by any method (including email) and encouraging electronic prescriptions [27,39]. Previous research [40] showed that less restrictive buprenorphine prescribing guidelines during COVID-19 led to improved retention in buprenorphine treatment for patients. However, our findings echoed those from Texas’s PDMP data [41], showing that COVID-19 still interrupted the upward trend of buprenorphine prescribers despite reduced restrictions on buprenorphine prescribing. In contrast to Texas [41], our examination of PDMP data did not reveal a significant declining trend in patient numbers or prescription volumes during the pandemic. A one-year study [42] following the pandemic from national claims data showed that the observed number of active buprenorphine episodes in December 2020 was comparable to the expected number. Our study extended the timeline to December 2021 and still found a remarkably different pattern in California. Post-COVID-19, buprenorphine prescriptions have seen a decrease from 53,407 per month to 46,855 per month. However, this declining trend did not reach statistical significance.

These insights from the most populous state contribute to understanding COVID-19’s impact on buprenorphine access. Encouragingly, our analysis demonstrates stability in patient and prescription volumes during the pandemic. The absence of detailed information in California’s CURES database constrains our capacity to thoroughly evaluate the impact of telehealth on buprenorphine dispensing. Being a retrospective study, it is susceptible to potential biases and confounding variables. Additionally, critical variables, such as the diagnosis of OUD, were unavailable in the databases and, consequently, were not included in this study. Lastly, the absence of data-sharing agreements among the nation’s PDMPs means that dispensing records of California residents cannot be tracked if they acquire prescriptions outside the state. While our dataset inherently comprises dispensing data from all residents of California, it is essential to note that the dispensing records may not necessarily reflect the actual usage of the medications.

Despite these limitations, our study is a pioneering effort highlighting the need to maintain or adjust policy strategies. On 29 December 2022, Congress eliminated the “Drug Addiction Treatment Act (DATA)-Waiver Program [43]” and allowed clinicians to treat patients with OUD without fear of special law enforcement activity, which may improve access to MOUDs. Future research could evaluate this change’s potential impact on buprenorphine prescribing practices.

## 5. Conclusions

Our study substantially expands the current understanding of buprenorphine prescribing patterns, encompassing periods both before and during the pandemic, with data analyzed up to the conclusion of 2021. Importantly, our examination of California’s PDMP data incorporates insights from uninsured patients, providing a comprehensive view of buprenorphine prescribing practices in the state. These findings enhance our knowledge of the influence of COVID-19 on buprenorphine access and emphasize the imperative need for policy adjustments to address the identified trends.

## Figures and Tables

**Figure 1 healthcare-12-00241-f001:**
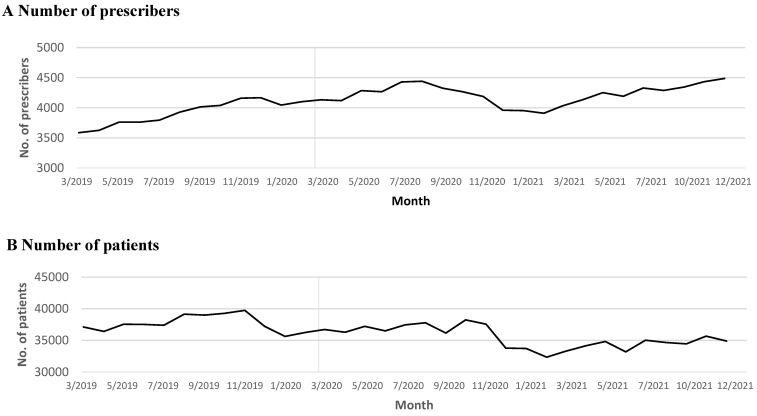
Monthly counts of patients, prescribers, and prescriptions from March 2019 to December 2021. The vertical line in the figures indicates the start of the COVID-19 pandemic.

**Table 1 healthcare-12-00241-t001:** Descriptive characteristics of buprenorphine dispensing before the pandemic and during the pandemic.

	Before Pandemic 19 March 2019–18 March 2020	During Pandemic 19 March 2020–18 December 2021
**Total refills**	640,883	983,961
**Average number of prescribers per month**	3916	4227
**Average number of patients per month**	37,679	35,410
**Prescriptions per month**	53,407	46,855
**Mean days of supply (day)**	20.67	22.10
**Mean daily dosage of each filling (mg)**	14.14	14.35
**Total number of patients**	92,723	126,957
**Total prescribers**	8166	11,590

**Table 2 healthcare-12-00241-t002:** Results of time series interrupted analysis for outcomes measures.

Outcomes	β Coefficient	*p* Value	95% Confidence Interval	
**Number of prescribers**				
Baseline trend (β_1_) before COVID-19	52.76	<0.001	38.92	66.60
Level change (β_2_) after COVID-19	−84.87	0.328	−259.35	89.60
The change in the trend (β_3_) after COVID-19	−47.41	<0.001	−64.50	−30.32
**Number of patients**				
Baseline trend (β_1_) before COVID-19	−8.60	0.939	−237.42	220.21
Level change (β_2_) after COVID-19	−600.25	0.585	−2820.80	1620.30
The change in the trend (β_3_) after COVID-19	−152.65	0.204	−392.99	87.69
**Number of prescriptions**				
Baseline trend(β_1_) before COVID-19	−143.16	0.376	−468.50	182.18
Level change (β_2_) after COVID-19	−3023.88	0.065	−6248.58	200.83
The change in the trend (β_3_) after COVID-19	−87.05	0.621	−443.13	269.03
**Daily dosage**				
Baseline trend (β_1_) before COVID-19	0.017	0.001	0.01	0.03
Level change (β_2_) after COVID-19	0.018	0.704	−0.08	0.11
The change in the trend (β_3_) after COVID-19	−0.009	0.063	−0.02	0.00
**Days of Supply**				
Baseline trend (β_1_) before COVID-19	0.02	0.148	−0.01	0.06
Level change (β_2_) after COVID-19	1.35	<0.001	1.00	1.69
The change in the trend (β_3_) after COVID-19	−0.03	0.083	−0.07	0.00

## Data Availability

The dataset was available upon the request at https://oag.ca.gov/cures.

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
