# Peer review of "Real-World Dispensing of Buprenorphine in California during Prepandemic and Pandemic Periods"

_healthcare, 2024, doi:10.3390/healthcare12020241_

Round 1

Reviewer 1 Report

Comments and Suggestions for Authors

The manuscript “Real-world dispensing of buprenorphine in California during Prepandemic and Pandemic Periods” addresses a timely and important topic, is generally well written, cautiously analyzed and interpreted.

General concerns:

A.      Line 27: “moderated the growth in prescribers” is technically correct but the audience has to work really hard to appreciate your strongest finding. Can you find another way to describe this in simpler terms?

B.      Supplemental Figures 1 and 2 are the foundation of this manuscript. These should be main figures and not supplemental. Having the date on the X-axis would be more informative though.

C.      Arguably, the tone of the discussion is so diplomatic as to be on the bland side. There is secret-shopper research (e.g. doi: 10.1016/j.jsat.2022.108788 or https://pubmed.ncbi.nlm.nih.gov/31158849/ ) that the vast majority of OUD patients nationally either can not access to a buprenorphine prescriber or can not do so in a timely manner. As the annual number of opioid overdoses seems to be either stable (at best) or increase, if the authors were able to be more forceful about the importance of the inability to reach patients with evidence-based OUD treatment, the urgency of this problem and need for more buprenorphine prescribers, that would increase the utility of this manuscript.

Minor points:

Line 3: A very light (friendly) suggestion. Pubmed indexes by middle-initials. Consider listing them when applicable.

L 28: Delete notably.

L 29: The conclusion should start with a re-statement of your most important finding.

The first paragraph provides a good overview of the nation but is a bit on the light side for California specific context. Where does CA rank relative to other states in opioid overdoses? This study (DOI: 10.1002/pds.4984 ) indicates that CA ranked 44th in the US (including Territories) for lowest buprenorphine distribution. Another (https://doi.org/10.1016/j.amepre.2018.01.034 Appendix 4) had CA at 44th for prescription opioids.

Methods: Consider adding a Procedures and Data-analysis sub-section headings.

L 103: Consider including a Supplementary Appendix that includes further information (e.g. brand names or NDC?) about the excluded formulations.

L 104: What % of the total prescription had this high dose ( > 50 mg) and were excluded?

L 127: The Tables should be free-standing so that a reader could just view the table and have a reasonable understanding of the findings. Adding “in California” and the data source to the legend would be helpful. Figure 1 and Table 1 legend should also provide more information. The X-axis on Fig 1 would be easier to read if the “20” in the year were deleted.

L 138: This reviewer has only done a couple of interrupted time series projects. Am not seeing the “Newer-West” standard errors.

L 157: IQVIA datasets usually have some gaps (VA patients, Indian Health Service patients-there are almost 100 reservations in CA, and non-coverage of some independent pharmacies). Consider explicitly noting how your PDMP study overcomes these data limitations.

L 213: “significantly” should be reserved for “statistically significant”. Consider “substantially” instead.

L 225 and 227: Delete “Number of” from Y-axis. Arguably, the text stating “Intervention starts: 13” is not needed and could be deleted.

Although a case-report, the untimely death of Matthew Perry in California from transdermal buprenorphine: https://news.yahoo.com/friends-co-creator-pays-tribute-215859305.html might be of interest. Is it known how often pain formulations are used off-label for OUD? Did this change during the pandemic? If there is anything published about this, this should be incorporated (albeit briefly) 

Author Response

We sincerely appreciate your comments. Please kindly help us find the response below.

A.      Line 27: “moderated the growth in prescribers” is technically correct but the audience has to work really hard to appreciate your strongest finding. Can you find another way to describe this in simpler terms?
We have changed it to “It impeded the upward trajectory of prescriber numbers that was evident prior to the onset of the pandemic”. Hope that improves the readability. 

B.      Supplemental Figures 1 and 2 are the foundation of this manuscript. These should be main figures and not supplemental. Having the date on the X-axis would be more informative though.
We have made remarkable changes to Table 2 and the sentences under “statistical analysis” so that the readers do not rely on the supplementary figures to understand our analysis. Thank you for pointing this out for us. 

C.      Arguably, the tone of the discussion is so diplomatic as to be on the bland side. There is secret-shopper research (e.g. doi: 10.1016/j.jsat.2022.108788 or https://pubmed.ncbi.nlm.nih.gov/31158849/ ) that the vast majority of OUD patients nationally either can not access to a buprenorphine prescriber or can not do so in a timely manner. As the annual number of opioid overdoses seems to be either stable (at best) or increase, if the authors were able to be more forceful about the importance of the inability to reach patients with evidence-based OUD treatment, the urgency of this problem and need for more buprenorphine prescribers, that would increase the utility of this manuscript.

The primary concern raised by the reviewer, namely, the limited access of the majority of opioid use disorder (OUD) patients to a buprenorphine prescriber, is largely attributed to the existence of the "X waiver/DATA waiver." Relevant literature supporting this viewpoint includes the work of Rowe et al. [1] and Gregory et al. [2].

[1]Rowe, Christopher L., Jennifer Ahern, Alan Hubbard, and Phillip O. Coffin. "Evaluating buprenorphine prescribing and opioid-related health outcomes following the expansion the buprenorphine waiver program." Journal of substance abuse treatment 132 (2022): 108452.
[2]Gregory, H.M., Hill, V.M., Parker, R.W. and Parker III, R.W., 2021. Implications of increased access to buprenorphine for medical providers in rural areas: a review of the literature and future directions. Cureus, 13(11).

Our paper, however, focuses on the evolution of buprenorphine prescribing patterns before and after the pandemic. As indicated in our concluding discussion paragraph, "On December 29, 2022, Congress eliminated the 'Drug Addiction Treatment Act (DATA)-Waiver Program[43],' allowing clinicians to treat patients with OUD without apprehension of special law enforcement activity, potentially enhancing access to Medications for Opioid Use Disorder (MOUDs). Future research could assess the impact of this legislative change on buprenorphine prescribing practices."

It is essential to note that, within the observational window of our study, no policy changes to the DATA waiver occurred, rendering this aspect beyond the scope of our investigation.

Line 3: A very light (friendly) suggestion. Pubmed indexes by middle-initials. Consider listing them when applicable.

We have updated the reference list.

L 28: Delete notably.
Thank you. We followed your comment and revised. 

L 29: The conclusion should start with a re-statement of your most important finding.
The first paragraph provides a good overview of the nation but is a bit on the light side for California specific context. Where does CA rank relative to other states in opioid overdoses? This study (DOI: 10.1002/pds.4984 ) indicates that CA ranked 44th in the US (including Territories) for lowest buprenorphine distribution. Another (https://doi.org/10.1016/j.amepre.2018.01.034 Appendix 4) had CA at 44th for prescription opioids.

We put our focus on the severity of the opioid crisis of the nation and our state. Although CA does not have a top ranking in opioid overdoses, 10,000 people died from it. Given that California is the most populous state, it's important to note that the ranking based on the percentage of the population overdosed may not accurately reflect the absence of an opioid overdose crisis in the state.

Methods: Consider adding a Procedures and Data-analysis sub-section headings.
Thank you, and the sub-section headings have been added.

L 103: Consider including a Supplementary Appendix that includes further information (e.g. brand names or NDC?) about the excluded formulations.

We acquired the CDC File of National Drug Codes for Selected Opioid Analgesics and Linked Oral Morphine Milligram Equivalent Conversion Factors, 2020 Version. However, we identified an issue with their list of NDC codes, as it proved incomplete, omitting more than 50% of the medications.  Due to the diverse nature of product names, we opted not to include an attached list. It's important to note that while our compilation may not be exhaustive to other states, it represents all the buprenorphine products preferred and prescribed by prescribers in California.

The original list of buprenorphine medications (both MOUD and analgesics): 
BUNAVAIL (film)
BELBUCA (film)
BUPRENORPHINE (patch)
BUPRENORPHINE HYDROCHLORIDE (powder/solution)
BUPRENORPHINE-NALOXONE (film)
BUPRENORPHINE-NALOXONE AVPAK (tablets)
BUTRANS (TDM)
SUBLOCADE 100MG(buprenorphine extended-release) , ERS
SUBLOCADE 300MG, ERS 
SUBOXONE (film)
SUBUTEX (tablet)
ZUBSOLV (tablet). 

We then excluded the formulations, such as extended-release solutions (ERS), implants, and powders. 

L 104: What % of the total prescription had this high dose ( > 50 mg) and were excluded?
Less than 1% of the prescriptions have been excluded. 

L 127: The Tables should be free-standing so that a reader could just view the table and have a reasonable understanding of the findings. Adding “in California” and the data source to the legend would be helpful. Figure 1 and Table 1 legend should also provide more information. The X-axis on Fig 1 would be easier to read if the “20” in the year were deleted.

We have modified the fonts of the figures and changed the legends in the tables.  

L 138: This reviewer has only done a couple of interrupted time series projects. Am not seeing the “Newer-West” standard errors.

Newey–West standard errors is used to handle autocorrelation in addition to possible heteroskedasticity. An assumption of standard regression models (Bernal JL et al) for interrupted time series regression is that observations are independent. This assumption is often violated in time series data because consecutive observations tend to be more similar to one another than those that are further apart, a phenomenon known as autocorrelation. However, we don’t think our data fits in the case of “autocorrelation”. 
Reference: Bernal JL, Cummins S, Gasparrini A. Interrupted time series regression for the evaluation of public health interventions: a tutorial. International journal of epidemiology. 2017 Feb 1;46(1):348-55.

L 157: IQVIA datasets usually have some gaps (VA patients, Indian Health Service patients-there are almost 100 reservations in CA, and non-coverage of some independent pharmacies). Consider explicitly noting how your PDMP study overcomes these data limitations.

We were unsure we understood the gaps the reviewer was referring to and the “100 reservations in CA”. We discussed this point in our discussion: "Compared to claims data, which exclusively encompass insured patients, our state-level PDMP database covered a broader spectrum of patients representing diverse insurance statuses. A survey focused on callers seeking buprenorphine treatment for OUD in several states highlighted that many buprenorphine prescribers do not accept Medicaid or un-insured patients [36]. In 2022, an estimated 3.2 million individuals in California remained uninsured [37]. The inclusiveness of Medicaid and uninsured patients in state-level PDMP data could facilitate understanding access to MOUD for all patients under multiple health insurance statutes and better inform policymakers. Our analysis provided a compre-hensive perspective on buprenorphine prescribing in California during the pandemic as the state PDMP program collected all the dispensing records of buprenorphine from all the pharmacies in California.”

L 213: “significantly” should be reserved for “statistically significant”. Consider “substantially” instead.
Thank you for the comments. We changed it. 

Although a case-report, the untimely death of Matthew Perry in California from transdermal buprenorphine: https://news.yahoo.com/friends-co-creator-pays-tribute-215859305.html might be of interest. Is it known how often pain formulations are used off-label for OUD? Did this change during the pandemic? If there is anything published about this, this should be incorporated (albeit briefly)

The comprehensive list of the prescribed  buprenorphine medication list in our PDMP data is

BUNAVAIL (film)
BELBUCA (film)
BUPRENORPHINE (patch)
BUPRENORPHINE HYDROCHLORIDE (powder/solution)
BUPRENORPHINE-NALOXONE (film)
BUPRENORPHINE-NALOXONE AVPAK (tablets)
BUTRANS (TDM)
SUBLOCADE 100MG(buprenorphine extended-release) , ERS
SUBLOCADE 300MG, ERS 
SUBOXONE (film)
SUBUTEX (tablet)
ZUBSOLV (tablet). 

We found it was not a challenging job to follow the indication on accessdata.fda.gov  to distinguish between buprenorphine for analgesics and MOUD. For example, 

BELBUCA (film): BELBUCA buccal film contains buprenorphine, a partial opioid agonist. BELBUCA is indicated for the management of pain severe enough to require daily, around-the-clock, long-term opioid treatment and for which alternative treatment options are inadequate.

BUNAVAIL (film): BUNAVAIL is a partial opioid agonist indicated for the maintenance
treatment of opioid dependence. Prescription use of this product is limited
under the Drug Addiction Treatment Act.

Off-label drug use involves prescribing medications for an indication, or using a dosage or dosage form, that has not been approved by the FDA. Our buprenorphine medication as MOUD was not a typical off-label use.

Reviewer 2 Report

Comments and Suggestions for Authors

Dear Authors,

I commend you for the time and effort invested in this work, including your mode of presentation. No doubt, opioid use disorder (OUD) is a major crisis in the US for which Buprenorphine is used as a medication-assisted treatment (MAT) for individuals with OUD. Choice of analytic method is appropriate.

Here are a few concerns and recommendations for your consideration:

Introduction: You started by talking about opioid overdose and then switched to Opioid use disorder (OUD) in a way that suggests that they are the same. Please note that they are not the same. Opioid overdose occurs when a person takes an excessive amount of opioids, leading to life-threatening respiratory depression (inadequate breathing) and potentially fatal consequences. In contrast, OUD is a chronic medical condition characterized by the compulsive use of opioids despite harmful consequences. Besides, OUD is a mental health condition (according to DSM-5). It may be worthwhile to present how opioid overdose relates to OUD in the introduction; otherwise, consider rephrasing the introduction to focus on OUD.

Method: You stated that you are using data from 2018 - 2021 (line 80) and mentioned in lines 93-4 that you are starting from March 2019. Consider reconciling the time differences.

The pre-pandemic period data used was 12 months, while the pandemic data was 21 months. Consider using 21 months of data for the pre-pandemic periods for equality and validity. Otherwise, provide a valid rationale for using 12 months. 

Results & Discussion: if you choose to consider 21 months pre-pandemic, I expect to see some changes in the results section. It would be interesting to see how this differs from what you have presented.

If you decide to stick with the 12 months pre-pandemic with valid rationale, it would be interesting to understand how this perceived unequal balance can be a limitation to the study's objectives.

Author Response

We appreciate your comments. Please kindly help us find our response to your comments below. 

Introduction: You started by talking about opioid overdose and then switched to Opioid use disorder (OUD) in a way that suggests that they are the same. Please note that they are not the same. Opioid overdose occurs when a person takes an excessive amount of opioids, leading to life-threatening respiratory depression (inadequate breathing) and potentially fatal consequences. In contrast, OUD is a chronic medical condition characterized by the compulsive use of opioids despite harmful consequences. Besides, OUD is a mental health condition (according to DSM-5). It may be worthwhile to present how opioid overdose relates to OUD in the introduction; otherwise, consider rephrasing the introduction to focus on OUD.

Thank you for your feedback. We have revised the introduction by reorganizing its structure. In the initial paragraph, we delve into the origins of the opioid crisis, emphasizing the significant role played by opioids, including prescription opioids. The subsequent paragraph commences with a crucial statistic: "Approximately 1 in 4 patients receiving opioid prescriptions for pain management develops opioid use disorder (OUD) [6]." It then highlights a source of optimism, stating, "The effective medications for opioid use disorder (MOUDs) are one reason for optimism [7,8]."

Method: You stated that you are using data from 2018 - 2021 (line 80) and mentioned in lines 93-4 that you are starting from March 2019. Consider reconciling the time differences.
The pre-pandemic period data used was 12 months, while the pandemic data was 21 months. Consider using 21 months of data for the pre-pandemic periods for equality and validity. Otherwise, provide a valid rationale for using 12 months. 
Results & Discussion: if you choose to consider 21 months pre-pandemic, I expect to see some changes in the results section. It would be interesting to see how this differs from what you have presented.
If you decide to stick with the 12 months pre-pandemic with valid rationale, it would be interesting to understand how this perceived unequal balance can be a limitation to the study's objectives.

We chose to use March 2019 because the PDMP in California was not mandated until the end of 2018. The previous years were a slow uptake period, which may not ensure the completeness of all the prescriptions dispensed in California. According to the MDRC Working Paper on Research Methodology, “The Validity and Precision of the Comparative Interrupted Time Series Design and the Difference-in-Difference Design in Educational Evaluation”, ITS design can be implemented with a minimum of four baseline time points (before the intervention). In our new methodology, “Segmented regression helps assess immediate (level) changes in the outcome rate and trend (slope).  In standard Interrupted Time Series analyses [1], the following segmented regression model is employed: Yt = β0  + β1T + β2Xt  +  β3TXt, where, β0 represents the baseline level at T=0, β1 is interpreted as the change in the outcome associated with a time unit increase (representing the underlying pre-intervention trend), β2 is the level change following the intervention, and β3 indicates the slope change following the intervention (utilizing the interaction between time and intervention TXt). We reported the significance level of β1 (baseline level), β2 (level change), and β3 (the slope change) coefficients in Table 2. A positive β1 value suggested an upward trend, while a negative value indicated the decline tendency.”
The prepandemic and post-pandemic periods do not have to be equal to generate meaningful estimates. 

Thank you once agin.

Reviewer 3 Report

Comments and Suggestions for Authors

Dear Authors,

 I read with great interest the manuscript entitled “Real World Dispensing of Buprenorphine in California During Prepandemic and Pandemic Periods", submitted to Healthcare Journal. The study is certainly engaging, exploring the dispensing trends of buprenorphine in California during distinct periods. The manuscript is well-written and easy to follow, yet I believe some enhancements could significantly elevate the clarity and depth of your work.

Methods Section (Page 3)

The authors state: “We analyzed data using Stata SE, version 17 (StataCorp LLC), with a predetermined significance level of 0.05.”

The current explanation of the statistical methods employed is brief and lacks clarity. A more comprehensive methods section detailing the specific statistical tests used, along with the rationale and significance level, is essential for readers to fully understand the analytical approach.

Table 1 (Page 3)

The authors describe:

- Average number of prescribers per month

- Average number of patients per month

- Prescriptions per month

- Mean days of supply (day)

- Mean daily dosage of each filling (mg)

The table presents valuable information, but lacks measures of statistical dispersion. Introducing standard deviations (SD) for variables and performing t-tests with p-values and 95% CI for the comparison between prepandemic and pandemic periods would provide a more comprehensive and interpretable presentation.

Maybe it will be interesting to add a “mean = prescription per month / patients per month” for each month and compare means between before pandemic vs. during pandemic (t-test).

Figure 1A, B, and C (Page 4)

Legends in the X-axis are challenging to read. Consider angling legends for better readability, and possibly adjust the size of the graphs to enhance clarity (i.e. make taller graphs).

The graphs show scatter plots with straight lines, which are not very informative. Please use scatter plots instead (without connecting line) to observe individual points. And maybe, for each graph and for both before pandemic and during pandemic time frames, it will be interesting to add a regression line, the equation for the regression, the correlation coefficient and the p-value. To assess differences between the two time frames, the authors can perform a test to compare slopes (covariates) and give and F-value and p-value. The authors should also check homoscedasticity in the regression, making a scatterplot with the residuals (which can be presented as suppl. material) and performing a t-test.

It can be interesting to check whether mean = prescription per month / patients per month” vs. month shows a positive correlation for both time frames. If so, it can be important to perform regression analysis and add regression lines, the equations, the correlation coefficients and the p-values, as well as to compare slopes (covariates) and give and F-value and p-value. (also, check homoscedasticity).

P.4 lines 140-142

Statements such as "there was no notable change in patient numbers or monthly prescription volumes (refer to Figure 1, Supplementary Figure 1, and Table 2).”" lack empirical evidence in the form of statistical comparisons. Performing tests to compare slopes (covariates) and providing F-values and p-values would substantiate these claims.

Table 2 (Page 4-5)

The legend requires more explicit details on the tests conducted and the significance levels.

Coefficient: please explain which coefficient

p value: please explain for which test. And if statistically significant, add asterisks in brackets (*) (**) (***) and highlight the p-value.

95% and Confidence Interval: It is not clear whether this is a single column (I assume so) or two different columns (it does not make sense to me). Please format table to amend it.

It seems that supplementary table 1 is exactly the same as table 2. Also, in suppl. table it is clear that 95% confidence Interval is a single column. Please address this question.

Discussion (Page 5)

The authors state: “Furthermore, there was a decline in the number of buprenorphine prescriptions dispensed, both in terms of quantity and growth rate, during the pandemic 34. They also identified an increase in the average per of days' supply of buprenorphine prescriptions 34.”

Proposing hypotheses for the decline in buprenorphine prescriptions and the increase in average days' supply during the pandemic could enhance the interpretability of your findings. Could one possible explanation be a change in the mortality rate? I think that the authors have not found an increase in the average per of days' supply of buprenorphine prescriptions.

The authors state: “The gradual increase in active buprenorphine prescribers in our observations was a positive sign before the pandemic. Previous literature suggested that buprenorphine prescriptions and prescribers increased by 36% and 86% from 2016 to 2021 nationally 37 . During the pandemic, the Drug Enforcement Administration temporarily relaxed outpatient buprenorphine prescribing regulations 27, 38 . Revisions included permitting prescribing to new patients via telephone or telemedicine and to existing patients by any method (including email) and encouraging electronic prescriptions 27, 38.”

It is clear that these data reflect prescriptions, not real intake. For example, a person can obtain prescribed buprenorphine, but maybe it can illicitly being sold. In this case, prescriptions can significantly differ from buprenorphine consumption. Consider addressing potential discrepancies between prescribed and consumed buprenorphine.

 Conclusion (Page 5)

“These findings ……………emphasize the imperative  need for policy adjustments to address the identified trends.”

The conclusion, emphasizing the need for policy adjustments, could benefit from a more explicit connection to the study's results. Clarify how the identified trends in dispensing support this imperative need for policy changes.

Supplementary files

Supplemental figures are indeed figure 1, but drawn as scatter plot. This, they are not informative, especially if figure 1 is modified accordingly to my suggestions. Also, as I said before, it seems that supplementary table 1 is exactly the same as table 2.

References

Several references lack crucial details such as access dates for websites. Please ensure the completeness of references, particularly for online sources.

This is the case of:

5. System NVS. Provisional Drug Overdose Death Counts2022.

6. California Department of Health Care Services. California response to the overdose crisis2022.

30. General SoCDoJOotA. Controlled Substance Utilization Review and Evaluation System2023.

It is not clear what kind of reference are the following, and whether an access date is needed:

20. Ghoshal M. How Race Affects Access to Opioid Use Disorder Medications.

36. Dietz M, Lucia L, Challenor T, et al. Undocumented californians projected to remain the largest group of uninsured in the state 313 in 2022. 2021.

42. Justice UDo. Informational Documents2023.

I hope you find these suggestions constructive, and I believe that these refinements will enhance the impact and clarity of your manuscript.

Author Response

We sincerely appreciate your comments. Please kindly help us find the reply below. 

1. The authors state: “We analyzed data using Stata SE, version 17 (StataCorp LLC), with a predetermined significance level of 0.05.”

The current explanation of the statistical methods employed is brief and lacks clarity. A more comprehensive methods section detailing the specific statistical tests used, along with the rationale and significance level, is essential for readers to fully understand the analytical approach.

Table 1 (Page 3)

The authors describe:

- Average number of prescribers per month

- Average number of patients per month

- Prescriptions per month

- Mean days of supply (day)

- Mean daily dosage of each filling (mg)

The table presents valuable information but lacks measures of statistical dispersion. Introducing standard deviations (SD) for variables and performing t-tests with p-values and 95% CI for the comparison between prepandemic and pandemic periods would provide a more comprehensive and interpretable presentation. Maybe it will be interesting to add a “mean = prescription per month / patients per month” for each month and compare means between before pandemic vs. during pandemic (t-test).

We appreciate your expertise. We have changed our methodology in the new submission:

“For our statistical analysis, we employed interrupted time series analysis, a widely recognized and robust approach to quantify changes in trends before and during the COVID-19 pandemic. Interrupted time series [33] is a potent quasi-experimental research design and a powerful tool for evaluating the impact of policy changes or quality improvement programs on the rate of a specific outcome in a defined population. The time series, which involves repeated observations of an event over time, is divided into two segments. The first segment encompasses rates of the event before the intervention or policy, marked in this case by the onset of the COVID-19 outbreak in March 2020, while the second segment represents rates after March 2020. “Segmented regression” is employed to statistically measure changes in level and slope during the post-pandemic period compared to the prepandemic period. The term “segmented” indicates a model with different intercept and slope coefficients for the pre- and post-pandemic time periods [33]. Segmented regression helps assess immediate (level) changes in the outcome rate and trend (slope).  In standard Interrupted Time Series analyses [34], the following seg-mented regression model is employed: Yt = β0  + β1T + β2Xt  +  β3TXt, where, β0 repre-sents the baseline level at T=0, β1 is interpreted as the change in the outcome associated with a time unit increase (representing the underlying pre-intervention trend), β2 is the level change following the intervention, and β3 indicates the slope change following the intervention (utilizing the interaction between time and intervention TXt). We reported the significance level of β1 (baseline level), β2 (level change), and β3 (the slope change) coefficients in Table 2.”

Figure 1A, B, and C (Page 4)

Legends in the X-axis are challenging to read. Consider angling legends for better readability, and possibly adjust the size of the graphs to enhance clarity (i.e. make taller graphs).

Thank you for pointing this out to us. We have changed the font of the legends for the x-axis to improve the readability.

The graphs show scatter plots with straight lines, which are not very informative. Please use scatter plots instead (without connecting line) to observe individual points. And maybe, for each graph and for both before pandemic and during pandemic time frames, it will be interesting to add a regression line, the equation for the regression, the correlation coefficient and the p-value. To assess differences between the two time frames, the authors can perform a test to compare slopes (covariates) and give and F-value and p-value. The authors should also check homoscedasticity in the regression, making a scatterplot with the residuals (which can be presented as suppl. material) and performing a t-test.

It can be interesting to check whether mean = prescription per month / patients per month” vs. month shows a positive correlation for both time frames. If so, it can be important to perform regression analysis and add regression lines, the equations, the correlation coefficients and the p-values, as well as to compare slopes (covariates) and give and F-value and p-value. (also, check homoscedasticity).

Table 2 (Page 4-5)

The legend requires more explicit details on the tests conducted and the significance levels.

Coefficient: please explain which coefficient

p value: please explain for which test. And if statistically significant, add asterisks in brackets (*) (**) (***) and highlight the p-value.

95% and Confidence Interval: It is not clear whether this is a single column (I assume so) or two different columns (it does not make sense to me). Please format table to amend it.

It seems that supplementary table 1 is exactly the same as table 2. Also, in suppl. table it is clear that 95% confidence Interval is a single column. Please address this question.

We appreciate the valuable comments on the statistical methodology. To address this, we have included the following sentences under the statistical methods: “In standard Interrupted Time Series analyses [34], the following segmented regression model is employed: Yt = β0  + β1T + β2Xt  +  β3TXt, where, β0 represents the baseline level at T=0, β1 is interpreted as the change in the outcome associated with a time unit increase (representing the underlying pre-intervention trend), β2 is the level change following the intervention, and β3 indicates the slope change following the intervention (utilizing the interaction between time and intervention TXt). We reported the significance level of β1 (baseline level), β2 (level change), and β3 (the slope change) coefficients in Table 2.”

Additionally, we have modified the first column of Table 2 to read as follows: “Baseline trend (β1) before COVID-19”, “Level change (β2) after COVID-19”, and “The change of the trend (β3) after COVID-19”.

In line with previous researchers [1], we have chosen to report the coefficient β. As suggested [1], “Change in slope and change in level were the most common intervention effects for time series interrupted analysis...75 (93%) of the literature reported an immediate change while 17 (21%) reported other changes in levels, for example, 12 months after the intervention”.

Reference

  1. Hudson, J.; Fielding, S.; Ramsay, C.R. Methodology and reporting characteristics of studies using interrupted time series design in healthcare. BMC medical research methodology 2019, 19, 1-7.

P.4 lines 140-142

Statements such as “there was no notable change in patient numbers or monthly prescription volumes (refer to Figure 1, Supplementary Figure 1, and Table 2).”” Lack of empirical evidence in the form of statistical comparisons. Performing tests to compare slopes (covariates) and providing F-values and p-values would substantiate these claims.

We demonstrated a nonsignificant p-value (p=0.204 and p=0.621 respectively in Table 2) in row 3, “The change of the trend (β3) after COVID-19” under the subtitles of “number of patients” and “number of prescriptions,” which is the empirical evidence of the form of statistical comparisons.

Discussion (Page 5)

The authors state: “Furthermore, there was a decline in the number of buprenorphine prescriptions dispensed, both in terms of quantity and growth rate, during the pandemic 34. They also identified an increase in the average per of days' supply of buprenorphine prescriptions 34.”

Proposing hypotheses for the decline in buprenorphine prescriptions and the increase in average days' supply during the pandemic could enhance the interpretability of your findings. Could one possible explanation be a change in the mortality rate? I think that the authors have not found an increase in the average per of days' supply of buprenorphine prescriptions.

In the sentences you mentioned, we cited the findings from IQVIA claims data. Subsequent to these statements, we juxtaposed our results with theirs by stating, 'Although our study in California revealed an erosion in the raw monthly count of patients filling buprenorphine prescriptions and a reduction in prescriptions, our findings indicated that these changes occurred without a noticeable alteration in the overall trend.

The authors state: “The gradual increase in active buprenorphine prescribers in our observations was a positive sign before the pandemic. Previous literature suggested that buprenorphine prescriptions and prescribers increased by 36% and 86% from 2016 to 2021 nationally 37 . During the pandemic, the Drug Enforcement Administration temporarily relaxed outpatient buprenorphine prescribing regulations 27, 38 . Revisions included permitting prescribing to new patients via telephone or telemedicine and to existing patients by any method (including email) and encouraging electronic prescriptions 27, 38.” It is clear that these data reflect prescriptions, not real intake. For example, a person can obtain prescribed buprenorphine, but maybe it can illicitly being sold. In this case, prescriptions can significantly differ from buprenorphine consumption. Consider addressing potential discrepancies between prescribed and consumed buprenorphine.

We agreed with you and added this point to our limitations “While our dataset inherently comprises dispensing data from all local residents of California, it is important to note that the dispensing records may not necessarily reflect their actual usage of the medications.”

 Conclusion (Page 5)

“These findings ……………emphasize the imperative  need for policy adjustments to address the identified trends.”

The conclusion, emphasizing the need for policy adjustments, could benefit from a more explicit connection to the study's results. Clarify how the identified trends in dispensing support this imperative need for policy changes.

In the preceding paragraph, we concluded “Despite these limitations, our study is a pioneering effort highlighting the need to maintain or adjust policy strategies. On December 29, 2022, Congress eliminated the “Drug Addiction Treatment Act (DATA)-Waiver Program[43]” and allowed clinicians to treat patients with OUD without fear of special law enforcement activity, which may improve access to MOUDs. Future research could evaluate this change's potential impact on buprenorphine prescribing practices”

This “policy adjustment” aligns with the considerations discussed in the preceding paragraph.

Supplemental figures are indeed figure 1, but drawn as scatter plot. This, they are not informative, especially if figure 1 is modified accordingly to my suggestions. Also, as I said before, it seems that supplementary table 1 is exactly the same as table 2.

We decided to withdraw the supplementary Table 1 to avoid confusion.   

References

Several references lack crucial details such as access dates for websites. Please ensure the completeness of references, particularly for online sources.

This is the case of:

  1. System NVS. Provisional Drug Overdose Death Counts2022.
  2. California Department of Health Care Services. California response to the overdose crisis2022.
  3. General SoCDoJOotA. Controlled Substance Utilization Review and Evaluation System2023.

It is not clear what kind of reference are the following, and whether an access date is needed:

  1. Ghoshal M. How Race Affects Access to Opioid Use Disorder Medications.
  2. Dietz M, Lucia L, Challenor T, et al. Undocumented californians projected to remain the largest group of uninsured in the state 313 in 2022. 2021.
  3. Justice UDo. Informational Documents2023.

We fixed the reference list. Thank you for your help!

Round 2

Reviewer 2 Report

Comments and Suggestions for Authors

Dear Authors,

Appreciate your responses and for including information to the manuscript to improve its quality. Well done.